# Photocatalytic $CO_2$ reduction with aminoanthraquinone organic dyes

Qinqin Lei[1,2], Huiqing Yuan[1,2], Jiehao Du[1], Mei Ming[1], Shuang Yang[1], Ya Chen[1], Jingxiang Lei[1] & Zhiji Han [1] ✉

The direct utilization of solar energy to convert $CO_2$ into renewable chemicals remains a challenge. One essential difficulty is the development of efficient and inexpensive light-absorbers. Here we show a series of aminoanthraquinone organic dyes to promote the efficiency for visible light-driven $CO_2$ reduction to CO when coupled with an Fe porphyrin catalyst. Importantly, high turnover numbers can be obtained for both the photosensitizer and the catalyst, which has not been achieved in current light-driven systems. Structure-function study performed with substituents having distinct electronic effects reveals that the built-in donor-acceptor property of the photosensitizer significantly promotes the photocatalytic activity. We anticipate this study gives insight into the continued development of advanced photocatalysts for solar energy conversion.

Light-driven reduction of $CO_2$ into value-added chemicals represents a sustainable way for the direct utilization of solar energy and conversion of greenhouse gas[1–3]. In an artificial photosynthetic (AP) scheme, a photosensitizer (PS) harvests the sunlight and transfers the energetic electron to a catalytic center which reduces $CO_2$[4–6]. In the past decades, both homogeneous and heterogeneous AP systems have been investigated extensively for photocatalytic $CO_2$ reduction[7,8]. However, the activity of current systems is still low for practical use. A frequent challenge is the development of highly active PSs that promote light-driven redox reactions. For the rational design of catalytic systems, molecular approaches have shown great advantages in unraveling factors that govern photocatalytic reactions. In this context, we report here a class of highly active organic PSs in precious metal-free systems for photocatalytic $CO_2$ reduction.

Noble-metal-based PSs have demonstrated high activity in photocatalytic $CO_2$ reduction[6,9–19]. Their long-lived excited states facilitate electron transfer from the excited state of the PS (PS*) to the catalyst in an oxidative quenching pathway. To provide a potentially widespread implementation, accelerating progress has been made in the development of inexpensive PSs to perform the same catalytic reaction[20]. Recently, PSs based on first-row transition metals such as Cu[21–24] and Zn[25] have been studied for light-driven $CO_2$ reduction, with turnover numbers (TONs) of 40–1566.

Due to being readily available in nature and because they are synthetically easy to functionalize, organic PSs are promising alternative light-absorbers for photocatalytic $CO_2$ reduction[26–28]. However, there are only a few reported organic PSs functioning in the visible region, and these systems usually have to perform with high PS concentrations due to their relatively low activity[29–43]. For example, 9-cyanoanthracene has been reported to give a turnover frequency (TOF) of -0.015 h$^{-1}$ (vs. PS) in a noble-metal-free system using Fe−tetraphenylporphyrin as the catalyst (TOF$_{Fe}$ -1.5 h$^{-1}$)[29]. Acriflavine was found to exhibit a TON$_{PS}$ of 5, when employed with a Co dipyridyl catalyst[30]. Purpurin, reported by Lau, Robert, and Chen groups, has shown activity for the reduction of $CO_2$ to CO with a series of Co, Fe, Ni polypyridyl, and Fe porphyrin catalysts[31–33], achieving an optimal TON$_{PS}$ of 1300[31]. Later, Robert et al. found that a phenoxazine-based organic PS promotes the reduction of $CO_2$ to CO and $CH_4$ with a total TON$_{PS}$ of ~2 in 102 h[34]. 2,4,5,6-tetrakis(carbazole-9-yl)-1,3-dicyanobenzene studied by Chao et al. gives a TON$_{PS}$ up to 1196 in CO generation using *mono−* and *bis−*terpydrine Fe catalysts[35–37]. Eosin Y, which was originally reported in photocatalytic $H_2$ production by the Eisenberg group[38–40], is also an active PS for $CO_2$ reduction to formate when using Ni pyridylthiolate catalysts, although it performs with considerably lower activity (TON$_{PS}$ = 28) than that of the catalyst (TON$_{Ni}$ = 14,000)[41]. Because achieving high activity for both the PS and

[1]MOE Key Laboratory of Bioinorganic and Synthetic Chemistry, School of Chemistry, Sun Yat-sen University, Guangzhou 510275, China. [2]These authors contributed equally: Qinqin Lei, Huiqing Yuan. ✉e-mail: hanzhiji@mail.sysu.edu.cn

the catalyst would be beneficial for developing versatile photocatalytic systems and applications in other relevant studies such as photoelectrochemical and supramolecular photocatalytic systems, this difficulty has led to assembling complicated molecular architectures with precious metals[44–49].

Several methods have been studied for improving the photocatalytic activity of organic PSs[28]. For example, the introduction of a heavy atom (such as Br or I) or a heteroatom (such as S or Se) to the xanthene-based dye has been found to facilitate intersystem crossing to generate a longer-lived $^3\pi\pi^*$ state, leading to improved activity in photocatalytic $H_2$ production[38,50]. Tuning the electron donors and acceptors in organic dyes help increasing the power conversion efficiencies of dye-sensitized solar cells[51,52]. We recently demonstrated that the coordination of polyhydroxy-anthraquinones to a redox active Cu center effectively enhanced the photocatalytic activity in both proton and $CO_2$ reductions[22,53]. In the present study, we report the application of simple yet more active aminoanthraquinone organic PSs **1**–**6** (Fig. 1) for visible light-driven reduction of $CO_2$ to CO. Different from previous systems, high TONs for both the PS and the catalyst can be realized. In addition, the systems contain no precious metal and use commercially available organic PSs. The photochemical steps and mechanism for $CO_2$ reduction have been studied in detail. Our structure-function study shows that the donor–$\pi$–acceptor design of the anthraquinone (AQ) unit through controlling electron substituents facilitates faster reductive quenching of the PS* and results in a much higher catalytic rate for $CO_2$ reduction.

## Results and discussion

### Absorption, emission, and electrochemistry of PSs

**1**–**6** display strong electronic absorption across the visible region in dimethylformamide (DMF). The maximum absorption bands ranging from 478 to 592 nm can be largely tuned by altering the substituents on the anthraquinone moiety (Fig. 2). The molar absorption coefficients ($0.68–1.58 \times 10^4 \, M^{-1} \, cm^{-1}$) were calculated from the linear relationship between the absorbance and the concentration (Table 1 and Supplementary Figs. 1–6). Upon irradiation with 365 nm light, these PSs produce intense red fluorescence at the 600–700 nm region with lifetimes ($\tau_0$) of 0.66–1.02 nanoseconds (Table 1 and Supplementary Fig. 7).

To further examine the organic dyes as potential PSs for photocatalytic reactions, electrochemical measurements were conducted (Table 1). Cyclic voltammograms (CVs) of **1**–**6** show two reduction events (Supplementary Fig. 9), with both reversible waves for **1**–**5**, whereas for PS **6**, a reversible and a second quasi-reversible reduction peaks were observed. The exact reduction potentials were measured by square wave voltammetry (SWV) (Table 1 and Supplementary Fig. 10). The photophysical and redox properties of the aminoanthraquinone dyes (Table 1 and Supplementary Table 1) thus suggest they serve as promising PSs for photocatalytic $CO_2$ reduction.

## Photocatalytic $CO_2$ reduction

The activity of $CO_2$ reduction by PSs **1**–**6** was studied in $CO_2$-saturated DMF solutions under irradiation with a white light-emitting diode (LED, $\lambda > 400$ nm, 100 mW/cm²). FeTDHPP (Fig. 1) was used as the $CO_2$ reduction catalyst, for the reason that it has been demonstrated to provide high activity in photocatalytic systems from our previous study[22]. 1,3-dimethyl-2-phenyl-2,3-dihydro-1*H*-benzo[*d*]imidazole (BIH) was used as the sacrificial donor to replace the oxidative half-reaction in the AP scheme. The gaseous products in the headspace were quantified in real time by gas chromatography (GC) equipped with a thermal conductivity detector (TCD) and a flame ionization detector (FID).

Figure 3a and Table 1 display the photocatalytic results of **1**–**6** under the same conditions (20 µM PS, 0.6 µM FeTDHPP, 60 mM BIH). The yield rate of CO is shown in Supplementary Table 2. In the series of experiments, CO is observed as the major product and the production of $H_2$ is significantly suppressed. PSs **1**–**5**, with amino and hydroxyl substituents on the AQ, exhibit generally high selectivity of CO (>99%), whereas PS **6** which contains a strongly acidic sulfonyl group gives a slightly lower selectivity of 98.5% ± 0.6%. Importantly, varying the functional groups on the AQ ring results in very different TONs of CO. The systems with the amino-substituted AQs (**1**–**3**) show $TON_{Fe}$ of 2395–3551 in 48 h. Under the same conditions but with an OH-substituted aminoanthraquinone (**4**), a considerably higher TON of 8360 ± 449 was obtained. The activity of the system can be further improved by including a heavy atom Br as the substituent at the 2–position (**5**), achieving a TON of 21,616 ± 2351 and a TOF of 4028 ± 669 mole CO/h per mole of catalyst. However, changing the OH group to a sulfonyl one (**6**) markedly decreases the light-driven activity (TON = 907 ± 154).

To investigate the optimal activity of the PS, photocatalytic experiments were performed at high concentrations of FeTDHPP (20 µM) and BIH (60 mM) (Table 1), where the activity is controlled by the [PS] (Supplementary Figs. 11–13). The PS **1** shows a $TON_{CO}$ (vs. PS) of 2011 ± 257 in 72 h. The NH₂- or SO₃H- substituted ones result in decrease in activity (TON = 482 ± 76 for **2**, 1523 ± 126 for **3**, 1183 ± 78 for **6**). Consistent with the results described above, the PSs **4** and **5** give much higher activity in the series, with TONs of 2849 ± 161 and 6012 ± 606, respectively.

The high activity of the PS and the catalyst, although obtained at different catalytic conditions, suggests that it may be possible to realize high activity for both the PS and the catalyst in one photocatalytic system. Indeed, when the experiment was performed under the same concentration of **5** and FeTDHPP (Fig. 3b and Supplementary Table 3), the system achieves a TON as high as 4978 ± 326 and a quantum efficiency of 11.1% ± 0.9% at 450 nm (based on two photons per CO), underscoring that both the light-harvesting and the $CO_2$ reducing processes are efficient in catalysis. The exceptional light-driven activity of the study is in contrast to those reported for other noble metal-free systems which usually show very different activity for the PS and the catalyst (Supplementary Table 4).

To study factors that influenced the light-driven stability, each component was added to the system when the rate of CO production was substantially decreased (Supplementary Fig. 14). Although BIH was nearly consumed in the conditions, addition of BIH to the system did not resume the activity (Supplementary Fig. 14a), which suggests decomposition of either the PS or the catalyst. When a mixture of PS and BIH was added, only ~5% activity was recovered (Supplementary Fig. 14d), indicating that most of the catalyst has decomposed. Indeed, with the addition of catalyst and BIH, CO production continued with a ~50% increase (Supplementary Fig. 14f). However, even though all three components were added, a similar level (~60%) of recovery was observed (Supplementary Fig. 14g), which is presumably due to light competition between the decomposed species and the added PS. These results thus indicate that the Fe porphyrin catalyst undergoes a

**Fig. 1 | Structure diagram.** Structures of PSs **1**–**6** and FeTDHPP in the study.

| | | |
|---|---|---|
| **1** | $R_1$ = H, | $R_2$ = H |
| **2** | $R_1$ = NH₂, | $R_2$ = H |
| **3** | $R_1$ = H, | $R_2$ = NH₂ |
| **4** | $R_1$ = H, | $R_2$ = OH |
| **5** | $R_1$ = Br, | $R_2$ = OH |
| **6** | $R_1$ = SO₃H, | $R_2$ = Br |

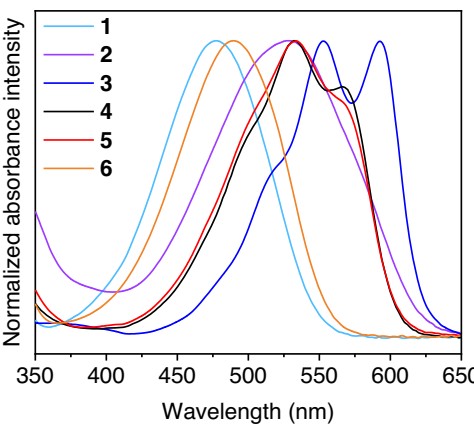

**Fig. 2 | UV–vis spectra.** Normalized absorption spectra of PSs **1**–**6** in DMF. Source data are provided as a Source Data file.

faster decomposition than the anthraquinone dye in the catalytic system.

The nature of the system was investigated. To confirm the homogeneity of the system, dynamic light scattering (DLS) and Hg-poisoning experiments were conducted. DLS results indicate there is no presence of nanoparticles in the pre- and post- catalytic systems (Supplementary Fig. 15). An excess amount of metallic $Hg^0$ in the system does not result in a significant change of the activity (Supplementary Fig. 16), which rules out the possibility that the activity of $CO_2$ reduction is contributed from amalgam-forming metal impurities. In addition, isotopic labeling experiments performed under an atmosphere of $^{13}CO_2$ show exclusive $^{13}CO$ as the product (Supplementary Fig. 17). These pieces of evidence are consistent with a homogeneous $CO_2$ reduction system in the study.

## Mechanism of $CO_2$ reduction

In a multi-component system, reductive quenching (electron transfer from the electron donor to the PS*) and oxidative quenching (electron transfer from the PS* to the catalyst) are two major photochemical pathways affecting the overall light-driven catalytic rate. Because aminoanthraquinone undergoes excited-state intramolecular proton transfer (ESIPT)[54], changing proton concentration may affect the fluorescence spectra during quenching experiments. In fact, we observed that the addition of BIH (which is slightly basic) to PSs **4** and **5** resulted in uneven quenching of the fluorescence at different wavelengths and that the fluorescence quenching rate constants ($k_q$) were calculated to be higher than the diffusion-controlled limit in DMF (Supplementary Figs. 18 and 19). Since the fluorescence lifetimes of the ESIPT tautomers have been reported to be identical[54], we determined the $k_q$ values by measuring the change of fluorescence lifetime in the presence of BIH. The reductive fluorescence quenching of **1**–**6** was found to be fast near the diffusion-controlled limit ($>10^9 M^{-1} s^{-1}$) (Table 1 and Supplementary Fig. 20). Because of significant overlap in both the absorption and the emission spectra of the PSs and the Fe catalyst (Supplementary Fig. 21), the oxidative quenching rate constants could not be obtained accurately. Based on the UV–vis and $^1H$ NMR spectra (Supplementary Figs. 22–29), there is no reaction between the PS and quenchers (BIH and FeTDHPP) at their ground states, which rules out a static quenching pathway. These results along with the fact that a much higher concentration of BIH (>3000 times) than that of the FeTDHPP in $CO_2$ reduction, suggests the system proceeds with a reductive quenching pathway (Fig. 4). However, since the triplet quantum yields of PSs **1** and **3** have been reported to be 3% and <0.1% in methanol respectively[55], it should be noted that reductive quenching occurring through $^3PS^*$ is another plausible

**Table 1 | Photophysical, electrochemical, and photocatalytic $CO_2$ reduction data of PSs 1–6**

| PS | λmax abs/nm (ε $M^{-1} cm^{-1}$) | λmax em (nm) | $E_{red}$ (V vs. SCE) | $TON_{Fe}^a$ | $TOF_{Fe}^a$ | $Sel_{CO}^a$ (%) | $CO^b$ (μmol) | $TON_{PS}^b$ | $Φ^c$ (%) | $Φ_{FL}$ (%) | $τ_o^d$ (ns) | $k_q^f$ ($M^{-1} s^{-1}$) |
|---|---|---|---|---|---|---|---|---|---|---|---|---|
| 1 | 478 (6790) | 600 | −0.96, −1.59 | 2395±228 | 1510±104 | 99.6±0.1 | 50±6 | 2011±257 | 8.9±0.8 | 4.7 | 0.88±0.013 | 3.9×10⁹ |
| 2 | 528 (8940) | 650 | −1.10, −1.70 | 2738±190 | 69±8 | 99.5±0.2 | 12±2 | 482±76 | 0.3±0.04 | 2.3 | 0.72±0.004 | 2.1×10⁹ |
| 3 | 592 (15,810) | 662 | −1.15, −1.64 | 3551±501 | 593±24 | 99.3±0.2 | 38±3 | 1523±126 | 3.0±0.1 | 5.3 | 0.82±0.003 | 2.7×10⁹ |
| 4 | 532 (12,040) | 620 | −0.84, −1.44 | 8360±449 | 1614±112 | 99.6±0.1 | 71±4 | 2849±161 | 8.1±0.3 | 7.1 | 0.94±0.008 | 5.2×10⁹ |
| 5 | 534 (9170) | 635 | −0.68, −1.19 | 21616±2351 | 4028±669 | >99.9 | 153±10 | 6012±606 | 11.1±0.9 | 7.0 | 1.02±0.005 | 7.5×10⁹ |
| 6 | 490 (7460) | 607 | −0.86, −1.30 | 907±154 | 93±17 | 98.5±0.6 | 30±2 | 1183±78 | 2.0±0.3 | 2.6 | 0.66±0.002 | 1.6×10⁹ |

Error bars denote standard deviations, based on at least three separated runs. Source data are provided as a Source Data file.
[a]60 mM BIH, 0.6 μM FeTDHPP, and 20 μM PS, λ > 400 nm, TON and Sel_CO calculated in 48 h, TOF calculated in 0.5 h for PS **1**, 2 h for PS **2**, and 1 h for PSs **3**–**6**.
[b]60 mM BIH, 20 μM FeTDHPP and 5 μM PS, λ > 400 nm, amount of CO and TON_PS calculated in 72 h.
[c]60 mM BIH, 20 μM FeTDHPP and 20 μM PS, λ = 450 nm, Φ calculated in 1 h.
[d]50 μM PS, a picosecond pulsed diode laser (λ = 472 nm) was used as the excitation source.
[e]Under $CO_2$. The λmax em of each photosensitizer is selected as the emission wavelength.
[f]$k_q$ calculated from linear fitting of the Stern–Volmer plot based on average values of three sets of data.

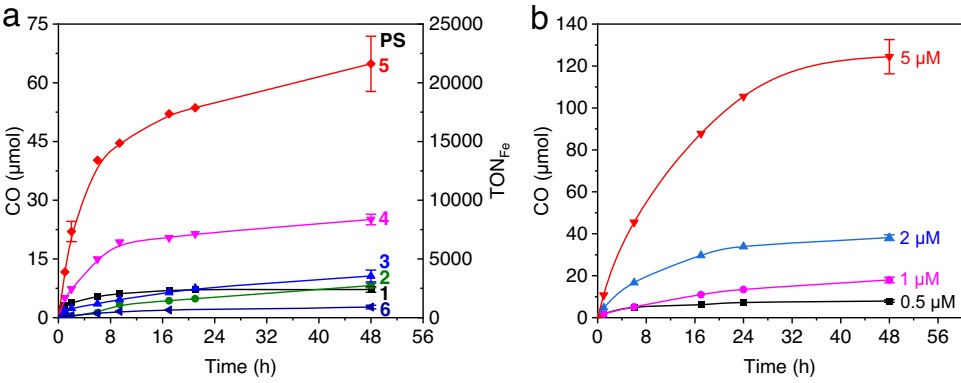

**Fig. 3 | Photocatalytic CO₂ reduction. a** Systems containing 20 μM **1–6**, 0.6 μM FeTDHPP, and 60 mM BIH; **b** systems containing 60 mM BIH, the same concentrations (0.5, 1, 2, 5 μM) of **5** and FeTDHPP. Irradiating conditions: CO₂ atmosphere,

5.0 mL CO₂-saturated DMF, 20 °C, white LED ($\lambda$ > 400 nm, 100 mW/cm²), 6.33 cm² light contact surface area. Error bars denote standard deviations, based on at least three separated runs. Source data are provided as a Source Data file.

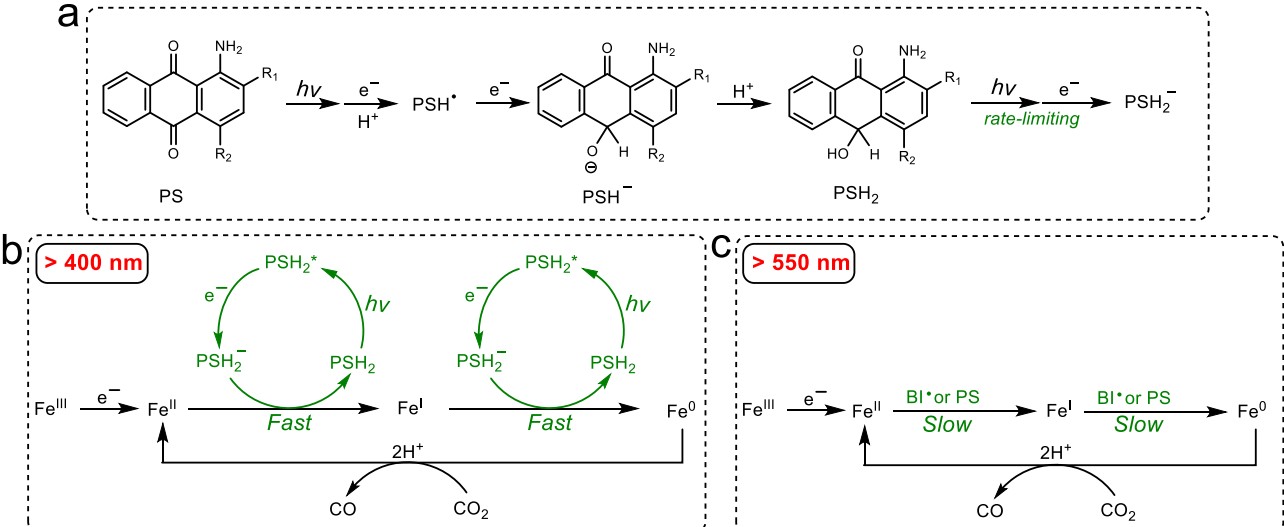

**Fig. 4 | Proposed mechanism of light-driven CO₂ reduction. a** Light-driven redox steps of the PS; **b** CO₂ reduction mechanism under >400 nm light irradiation; **c** CO₂ reduction mechanism under >550 nm light irradiation.

photochemical pathway, especially for the heavy-atom containing compound **5**. In fact, reductive quenching has been reported as the most common mechanism in other organic dye-containing systems[28,29,31,34,37]. Thus, the reductive rate and the reduction power of the reduced PS are two critical factors influencing the light-driven electron-transfer process in such photocatalytic reactions.

The light-driven redox process of the AQ-based dyes has been a subject of great interest in photochemistry[56–60], however, due to the various intermediates in reductions and protonations, the reaction mechanism is still under debate. Because AQ generates a similar intermediate at ~400 nm as PSs **1–6** during photolysis (Supplementary Figs. 30–36), the light-driven process of AQ was investigated to gain useful mechanistic information. UV–vis spectra reveal that photolysis of AQ in the presence of BIH in DMF quickly generates a species at 560 nm in 1 min (Fig. 5a). Since it is clear neither the 1 electron reduced AQ (545 nm) nor the 2 electron reduction product AQ²⁻ (622 nm) gives a good match[56], this species is assigned to an e⁻/H⁺ product AQH•. In fact, a semiquinone at 570 nm has been detected in an osmium triad[61,62]. An AQ⁻ species can be observed unambiguously from a reaction with NaBH₄ (Supplementary Fig. 37). It is not surprising that the AQ⁻[56] undergoes a fast

protonation by BIH⁺ to produce the AQH•. Upon continued irradiation, an intermediate at ~520 nm is observed (Fig. 5a), which is consistent with the generation of an AQH⁻[56]. A further protonation of the AQH⁻ species to generate a 10-hydroxyanthrone (AQH₂) is expected to take place based on ¹H NMR spectra (Supplementary Fig. 38). The UV–vis spectra also shows another absorption peak at ~407 nm (Fig. 5a), which is similar to a proposed AQH₂ intermediate at 407 nm from an AQ-containing pentad complex reported by Wenger et al.[59].

Robert et al. have shown that CO₂ reduction by FeTDHPP occurs at an Fe(0) oxidation state at −1.55 V vs. SCE[16,63,64]. However, our electrochemical studies reveal much positive reduction potentials for the PS⁻ and PS²⁻ species (Table 1), indicating that electron-transfer from these reduced species including their protonated forms PSH and PSH₂ (presumably with more positive potentials)[65,66] to the catalyst is unfavorable. The generation of the Fe(0) must arise from a more reduced PS. Indeed, the AQH₂ species generated from subsequent reductions and protonations gives a relatively long-lived (17.3 ns) fluorescence (Supplementary Figs. 39 and 40), which can be quickly quenched by BIH (Supplementary Fig. 41). Moreover, the AQH₂ moiety in a pentad system has been reported to be highly fluorescent with a long lifetime of 4.7 μs[59]. Thus, a plausible photochemical pathway involves

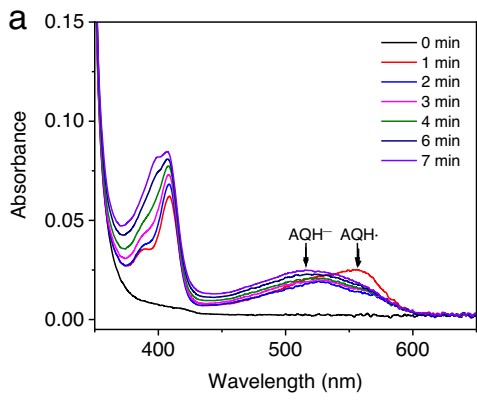
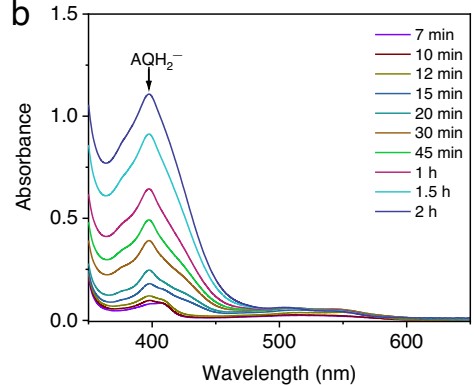

**Fig. 5 | UV–vis absorption spectra of AQ.** A system containing 0.02 mM AQ, 3.0 mM BIH, and 2 mL DMF in a quartz cuvette (10-mm path length) under $N_2$ upon irradiation with white LED light ($\lambda > 400$ nm, 100 mW/cm²) at 20 °C. Irradiation time ranges from 0 to 7 min (**a**), and from 7 min to 2 h (**b**). Source data are provided as a Source Data file.

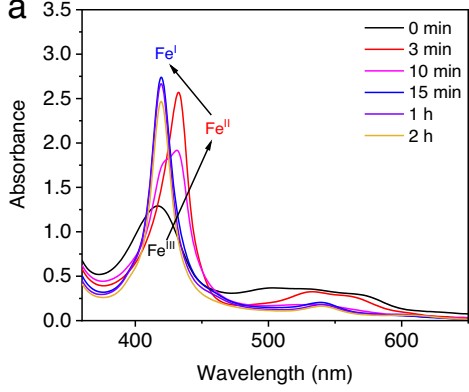
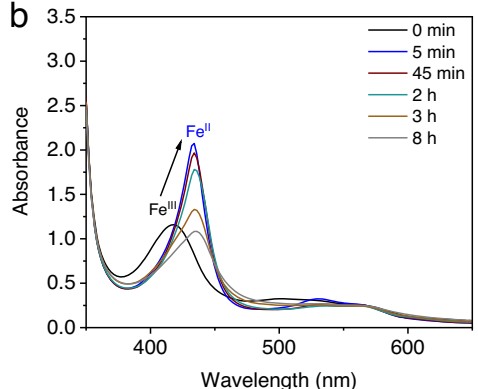

**Fig. 6 | UV–vis spectra investigating the reduction of FeTDHPP.** Systems containing 30 mM BIH, 20 μM FeTDHPP, 20 μM **5**, and 2 mL DMF in a quartz cuvette (10-mm path length) under $N_2$ at 20 °C, upon irradiation with (**a**) white LED light ($\lambda > 400$ nm, 100 mW/cm²), or with (**b**) a 300 W xenon light source equipped with a 550 nm cut-off filter ($\lambda > 550$ nm) for 8 h. Source data are provided as a Source Data file.

reduction of the Fe catalyst by a $PSH_2^-$ photoproduct, which can be generated from excitation of the $PSH_2$ followed by reductive quenching (Fig. 4).

The intermediate generated from AQ at ~400 nm starts to appear at 15 min and continue to increase in 2 h during photolysis (Fig. 5b), which is commonly observed with PSs **1**–**6** during $CO_2$ reduction (Supplementary Figs. 31–36). Although this intermediate exhibits a similar absorption feature as the $AQH_2$, a much slower generation of the species than that for the AQH• suggests it is not an $AQH_2$. Furthermore, we observed fast generation of CO and an Fe(I) species by adding 0.25 equiv of FeTDHPP (with respect to PS) to the 406 nm species generated from light irradiation of a mixture of **5** and BIH (Supplementary Fig. 42). In the experiments, an average of 0.34 equiv of CO (vs. PS) was obtained, which is close to the theoretical maximum yield (0.33 equiv) based on the proposed mechanism in Fig. 4. In a control experiment before generating the 406 nm species, no CO was detected (Supplementary Fig. 43). Based on these results, this photoproduct at ~400 nm is tentatively assigned to a $PSH_2^-$.

To examine the reduction power of $PSH_2^-$, in situ electrochemical measurements were conducted for the light-driven systems. SWV experiments with the photochemically generated species at ~400 nm show the appearance of new reduction waves at potentials more negative than −1.90 V vs. SCE for PSs **1**–**6** (Supplementary Fig. 44). Hence, electron-transfer from the $PSH_2^-$ to the FeTDHPP that leads to production of the required Fe(0) intermediate for $CO_2$ reduction is thermodynamically feasible.

Additional experiments were conducted to investigate reductions of the Fe catalyst. In the photocatalytic experiments with white LED ($\lambda > 400$ nm), UV–vis spectra suggest that the Fe(III) compound (416 nm) is completely converted to an Fe(II) species (432 nm) within 3 min and then to an Fe(I) species (420 nm) which continues to decrease during $CO_2$ reduction (Fig. 6a and Supplementary Fig. 45). This observation is consistent with a previously reported mechanism by Robert et al.[4,67]. Because both the $PSH_2^-$ and the BI• (−1.60 V vs. SCE in DMF)[22] are potential reductants in generating the Fe(I) and Fe(0) intermediates, it is crucial to understand the role of BIH in the system. In a photocatalytic experiment with 10 mM [BIH], the total amount of CO generated is near the theoretical maximum yield of BIH (Supplementary Fig. 46), which indicates BIH donates two electrons in $CO_2$ reduction. The first electron-transfer process is usually from BIH to the excited state of PS, which has been well-studied[68]. However, the actual mechanism by which the second electron of the sacrificial donor transfers from the significantly more reducing BI• to either the excited PS, or an Fe(II), or an Fe(I), remains uncertain.

To study this further, photocatalytic $CO_2$ reduction was conducted under a xenon light source equipped with a 550 nm cut-off filter to shut down the pathway involving the $PSH_2^-$. In this case, there is no $PSH_2^-$ observed from the UV–vis spectra (Supplementary Fig. 47), and CO production is considerably slower (TON = 6 in 5 h). The major catalytic species detected corresponds to the Fe(II), while the Fe(I) intermediate is present in a notably less amount compared with that generated under the $\lambda > 400$ nm light (Fig. 6b and Supplementary Figs. 47 and 48). Similar results are observed in experiments

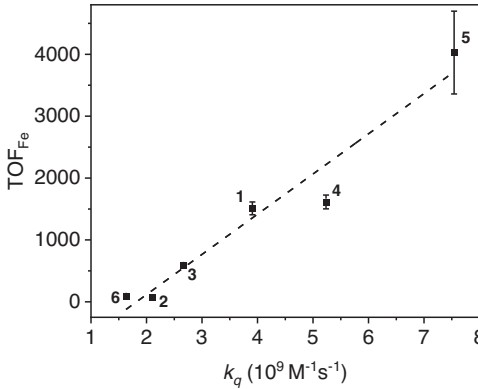

**Fig. 7 | Relationship of TOF and $k_q$.** Plot of the initial rates of CO generation with respect to the reductive quenching rate constants of PSs 1–6. The dotted line is shown for viewing convenience. Error bars denote standard deviations, based on at least three separated runs. Source data are provided as a Source Data file.

performed under 450 nm and 525 nm LED light with similar photon numbers (Supplementary Figs. 49 and 50). Meanwhile, the production of CO under 525 nm LED light (TON = 1.4 in 8 h) is also lower than that under the 450 nm LED light (TON = 344 in 8 h). Furthermore, irradiating a mixture of BIH and FeTDHPP gave almost no Fe(I) and only the Fe(II) was detected (Supplementary Fig. 51). These results are all consistent with the fact that although the BI• can reduce Fe(II) and Fe(I) to the Fe(0) intermediate, these are much slower processes compared with the ones using $PSH_2^-$ as the reductant. Thus, BI• is likely responsible for the reduction of $PSH_2^*$ in a photochemical step, while the resulting $PSH_2^-$ proceeds in reducing the Fe(II) or Fe(I) (Fig. 4).

The observation of the $PSH_2^-$ species by UV–vis during CO production suggests that the formation of $PSH_2^-$ may be rate-limiting in catalysis. Interestingly, we found that the TOF of CO production and the first reductive quenching $k_q$ of PSs 1–6 follows a generally linear trend, in which a faster quenching rate is observed with a higher TOF (Fig. 7). However, further evidence is necessary to identify the substituent effect on the photo-conversion of $PSH_2$ to $PSH_2^-$ in $CO_2$ reduction. In addition, no CO was detected from a system using AQ as the PS although an $AQH_2^-$ species was observed (Supplementary Fig. 30), which indicates the functional groups of 1–6 play important roles in promoting the generation of the $PSH_2^-$. Based on the electrochemical data of 1–6 and 1-amino-2-methylanthraquinone (Table 1 and Supplementary Figs. 9 and 10), the −NH₂ group acts as an electron-donating group while the −OH group is electron-withdrawing on AQ. These distinct electronic groups may essentially create an internal donor-acceptor property by de-symmetrizing the organic molecule, which facilitates electron-transfer of the dye. Consistent with the proposal, introducing −NH₂ to the 1−position of AQ (PS 1) greatly promotes photocatalytic $CO_2$ reduction, while having an additional −NH₂ group at the 4−position (PS 3) results in a decrease of activity. Furthermore, the higher activity of 4 and 5 may be attributed to the reverse electronic effects of the −OH and −Br groups as compared to the −NH₂. For PS 6, although sulfonyl is also electron-withdrawing, its activity of $CO_2$ reduction is considerably lower than that of 1–5. This is presumably due to that the acidic sulfonyl group promotes the transfer of proton to the catalyst thus enhancing $H_2$ generation. Indeed, the selectivity for $H_2$ of 6 is the highest in our study (Table 1). Although this selectivity (1.5%) is low compared with that of CO (98.5%), the intermediates generated from the $H_2$ pathway may significantly alter the reaction mechanism by introducing extra energy barriers in $CO_2$ reduction[4].

In summary, this paper describes the application of a series of simple organic light absorbers, based on naturally abundant anthraquinone dyes, in promoting visible light-driven $CO_2$ reduction. Unlike

previously reported systems, high activity for both the PS and the catalyst has been demonstrated in our study. The mechanistic study suggests that the hydroxyanthrone forms of PS ($PSH_2$ and $PSH_2^-$), generated from reductive quenching, are important intermediates in the light-driven catalytic steps. The most active PS was found by employing both electron-donating and withdrawing groups on the anthraquinone. Thus, this work presents a class of inexpensive dyes to access high activity in $CO_2$ reduction and provides understanding for improving other light-driven and light-electricity-driven systems for practical applications, such as water splitting, solar cell, and organic synthesis.

## Methods
### Materials
Compound 1 (97%) and compound 3 (>97%) were purchased from Aladdin. Compound 2 was purchased from Sigma-Aldrich. Compound 4 (96%) was purchased from Alfa Aesar. Compound 5 (97%) was purchased from BIDE. Compound 6 (98%) was purchased from Macklin. 1-Amino-2-methylanthraquinone (>90%) was purchased from Shanghai Xian Ding Biotechnology Co. Ltd. (Shanghai, China). Compound 1 was recrystallized twice from hot acetone. Compound 4 was recrystallized twice from a mixture of hot acetone and acetonitrile. 1-Amino-2-methylanthraquinone was recrystallized twice from hot ethanol until it is pure according to the ¹H NMR spectrum. Other solvents and chemicals are commercially purchased and used as obtained without further purification. BIH was prepared based on a method from the literature[69].

### Synthesis of FeTDHPP
FeTDHPP was prepared from a modified method of the literature[64]. A solution of 2′,6′-dimethoxybenzaldehyde (1.0 g, 6.02 mmol) and pyrrole (0.419 mL, 602 mmol) in CHCl₃ (600 mL) was degassed by N₂ for at least 20 min. BF₃·OEt₂ (0.228 mL, 0.87 mmol) was added drop by drop via a syringe. After the solution was stirred at room temperature under N₂ in the dark for 1.5 h, 2,3-dichloro-5,6-dicyano-1,4-benzoquinone (DDQ) (1.02 g, 4.51 mmol) was added. The mixture was stirred for an additional 1.5 h at reflux. After cooling to room temperature, the mixture was added with 1 mL of triethylamine to neutralize the excessive acid. Then the solvent was removed, and the resulting black solid was purified by column chromatography (silica gel, CH₂Cl₂) affording 5, 10, 15, 20-tetrakis(2′, 6′-dimethoxyphenyl)-21H,23H-porphyrin as a purple powder (290 mg, 23%). ¹H NMR (400 MHz, CDCl₃): δ 8.67 (s, 8H), 7.68 (t, J = 8.4 Hz, 4H), 6.98 (d, J = 8.4 Hz, 8H), 3.50 (s, 24H), −2.50 (s, 2H). HRMS (m/z): [M + H]⁺ calcd for C₅₂H₄₇N₄O₈ 855.33884; found, 855.33582. To a solution of 5, 10, 15, 20-tetrakis(2′,6′-dimethoxyphenyl)-21H,23H-porphyrin (200 mg, 0.235 mmol) in dry CH₂Cl₂ (10 mL) was added with BBr₃ (1.0 mL, 10.38 mmol) at 0 °C under N₂. The resulting green solution was allowed to stir for 24 h at room temperature. Then 4.0 mL of water was added at 0 °C and the mixture was stirred for 40 min. A saturated NaHCO₃ solution was added until the pH of the aqueous layer was around 7. Ethyl acetate (20 mL) was added to the suspension. The organic layer was separated, washed twice with water (20 mL), and then dried over anhydrous Na₂SO₄. The solvent was removed and the residue was purified by column chromatography (silica gel, 2:1 ethyl acetate/dichloromethane) to yield 5, 10, 15, 20-tetrakis(2′,6′-dihydroxyphenyl)-21H,23H-porphyrin as a purple powder (150 mg, 87%). ¹H NMR (400 MHz, MeOD): δ 8.92 (s, 8H), 7.50 (t, J = 8.2 Hz, 4H), 6.84 (d, J = 8.2 Hz, 8H). HRMS (m/z): [M + H]⁺ calcd for C₄₄H₃₁N₄O₈ 743.21364; found, 743.21204. FeTDHPP was prepared by heating a dry methanol solution containing 5, 10, 15, 20-tetrakis(2′,6′-dihydroxyphenyl)-21H,23H-porphyrin (100 mg, 0.135 mmol), FeCl₂·4H₂O (270 mg, 1.35 mmol), and 2,6-lutidine (39 μL, 0.335 mmol) at 50 °C for 3 h under N₂. After the solvent was removed, the resulting brown solid was dissolved in ethyl acetate (40 mL), added with 1.2 M HCl (40 mL), and stirred for 1 h. The organic layer was separated and washed several

times with saturated NaCl solution until the pH was neutral. The organic solvent was removed and the crude product was purified by column chromatography (silica gel, ethyl acetate) to give FeTDHPP as a brown solid (100 mg, 89%). HRMS ($m/z$): [M-Cl]$^+$ calcd for $C_{44}H_{28}FeN_4O_8$ 796.12511; found, 796.12343.

## Characterization

$^1$H NMR and $^{13}$C NMR spectra were recorded at a Bruker advance III 400-MHz NMR instrument (Supplementary Figs. 52–65). UV–vis spectra were recorded on a Thermo Scientific GENESYS 50 UV–visible spectrophotometer. The FT–IR spectra were recorded using a Nicolet/Nexus-670 FT–IR spectrometer (ATR mode) (Supplementary Fig. 66). HRMS spectra were obtained on a Thermo Fisher Scientific Orbitrap Q Exactive ion trap mass and Thermo Fisher Scientific LTQ Orbitrap Elite (Supplementary Figs. 67–72). Dynamic light-scattering experiments were tested with a Brookhaven Elite Sizer zata-potential and a particle-size analyzer. GC/MS experiments were performed with an Agilent 7890A-5975C instrument.

## Fluorescence quenching rate constant determination

A PS in DMF was degassed by $N_2$ or $CO_2$ for 15 min in a sealed quartz cuvette with a septum cap. Different from other PSs, the absorption spectrum of PS **5** changes slightly under $N_2$ when BIH was added (Supplementary Fig. 22). Therefore, the fluorescence quenching experiments for PS **5** were carried out under $CO_2$ and those for the rest of the PSs were performed under $N_2$. An identical excited-state lifetime for the PS was found either under $N_2$ or $CO_2$ (Supplementary Fig. 73). During the experiments, different concentrations of BIH were added to the solution of PS under $N_2$ or $CO_2$. The steady-state fluorescence for solution samples was measured by Duetta fluorescence and absorbance spectrometer. The excited-state lifetime of the photosensitizer was measured with an FLS 980 fluorescence spectrometer (Edinburgh instruments), in which a picosecond pulsed diode laser ($\lambda = 472$ and 406.2 nm) (Edinburgh instruments EPL-470) was used as the excitation source. The $\lambda_{max}$ of emission for each photosensitizer is selected as the emission wavelength. The instrumental response function (IRF) of the instrument was measured using silicon oxide (30% in $H_2O$) (Supplementary Fig. 74). The $k_q$ was calculated by the Stern–Volmer equation:

$$I_0/I \text{ or } \tau_0/\tau = 1 + k_q \times \tau_0 \times [Q] \qquad (1)$$

where $I_O$ and $I$ represent the fluorescence intensity of the photosensitizer in the absence and presence of a quencher; $\tau_O$ and $\tau$ is the lifetime of the photosensitizer in the absence and presence of the quencher; $k_q$ is the quenching rate constant; $[Q]$ is the concentration of the quencher BIH.

## Electrochemical measurements

Electrochemical studies were performed using a CHI-760E electrochemical analyzer using a single-compartment cell with a glassy carbon working electrode (3.0 mm in diameter), a platinum auxiliary electrode, and a SCE reference electrode. The electrolyte solution was 0.1 M tetrabutyl hexafluoroammonium phosphate in DMF. The solution was purged with $N_2$ or $CO_2$ for at least 30 min before measurement. All potentials reported in this study were referred to SCE.

## Photocatalytic CO₂ reduction

Photocatalytic experiments were conducted in a closed scintillation vial with rubber plug and magnetic stirring. The headspace of the vial was 51.8 mL. A reaction mixture (5.0 mL) was bubbled with $CO_2$ for 25 min and then irradiated with a LED light setup ($\lambda > 400$ nm, or $\lambda = 450$ nm, or $\lambda = 525$ nm, PCX-50 C, Beijing Perfectlight Technology Co., Ltd.) or a 300 W Xe lamp (PLS-SXE-300, Beijing Perfect light) equipped with a 550 nm cut-off filter. The gaseous products were analyzed by Shimadzu GC-2014 gas chromatography equipped with a

Shimadzu Molecular Sieve $13 \times 80/100$ $3.2 \times 2.1$ mm $\times 3.0$ m and a Porapak N $3.2 \times 2.1$ mm $\times 2.0$ m columns. A thermal conductivity detector (TCD) was used to detect $H_2$ and a flame ionization detector (FID) with a methanizer was used to detect CO and other hydrocarbons. Nitrogen was used as the carrier gas. The oven temperature was kept at 60 °C. The TCD detector and injection port were kept at 100 °C and 200 °C, respectively. Specifically, systems containing 60 mM BIH (0.3 mmol, 67.2 mg), 0.6 µM FeTDHPP (0.003 µmol, 2.5 µg) and 20 µM PS (0.1 µmol, 22.3 µg **1**; 23.8 µg **2**; 23.8 µg **3**; 23.9 µg **4**; 31.8 µg **5**; and 38.2 µg **6**) were used for the calculation of $TON_{Fe}$, $TOF_{Fe}$ and $Sel_{CO}$. Systems containing 60 mM BIH (0.3 mmol, 67.2 mg), 20 µM FeTDHPP (0.1 µmol, 83.2 µg) and 5 µM PS (0.025 µmol, 5.6 µg **1**; 6.0 µg **2**; 6.0 µg **3**; 6.0 µg **4**; 8.0 µg **5**; and 9.6 µg **6**) were used for the calculation of the yield of CO and $TON_{PS}$.

## Quantum yield of CO production

The experiments were carried out under monochromic light of 450 nm obtained using a blue LED light setup ($\lambda = 450$ nm, PCX-50C, Beijing Perfectlight Technology Co., Ltd.). The blank was a DMF solution containing 60 mM BIH and 20 µM FeTDHPP. Systems containing 60 mM BIH (0.3 mmol, 67.2 mg), 20 µM FeTDHPP (0.1 µmol, 83.2 µg) and 20 µM PS (0.1 µmol, 22.3 µg **1**; 23.8 µg **2**; 23.8 µg **3**; 23.9 µg **4**; 31.8 µg **5**; and 38.2 µg **6**) were used for the calculation of the quantum yields. The difference between the power ($P$) of light passing through the blank and through the sample containing the photosensitizer was measured by a FZ-A Power meter (Beijing Normal University Optical Instrument Company). The quantum yield ($\Phi$) was calculated after 1 h irradiation according to the following equation:

$$\Phi = \frac{2 \times n(CO) \times N_A}{PSt \times \frac{\lambda}{hc}} \qquad (2)$$

where $n$ (CO) is the number of CO molecules produced, $N_A$ is the Avogadro constant ($6.02 \times 10^{23}$ mol$^{-1}$), $S$ is the incident irradiation area (6.33 cm$^2$), $t$ is the irradiation time (in second), $\lambda$ is the incident wavelength (450 nm), $h$ is the Plank constant ($6.626 \times 10^{-34}$ J s), and $c$ is the speed of light ($3.0 \times 10^8$ m s$^{-1}$).

## Fluorescence quantum yield

Fluorescence quantum yields of PSs (listed in Table 1) were calculated according to a literature method[50]. A PS was dissolved in DMF and bubbled with $N_2$ for at least 10 min in a quartz cuvette (10-mm path length). Rhodamine 6G (R6G) was used as a standard sample. UV–vis spectra (Supplementary Fig. 75) were measured with Thermo Scientific GENESYS 50 UV–visible spectrophotometer and emission spectra (Supplementary Fig. 76) were acquired on a Duetta fluorescence and absorbance spectrometer. The absorption of photosensitizers were adjusted to the same as that of R6G at $\lambda = 480$ nm. The fluorescence quantum efficiency was calculated as follow:

$$\Phi_x = \Phi_{st} \left( \frac{Abs_{st}}{Abs_x} \right) \left( \frac{\eta_x^2}{\eta_{st}^2} \right) \left( \frac{Area_{Emx}}{Area_{Emst}} \right) \qquad (3)$$

where $\Phi_{st}$ is the fluorescence quantum yield of R6G ($\Phi_{st} = 0.95$ in EtOH); $\eta$ is the refractive index of solvent ($\eta_{EtOH} = 1.3611$, $\eta_{DMF} = 1.4300$); $Area_{Em}$ is the emission integral area of the photosensitizer or R6G.

## Statistics and reproducibility

The statistical analysis is based on the original data without randomization and blind treatment. In order to ensure the reproducibility of the data, key experiments were conducted at least three separated runs with freshly prepared solutions on different days.

## Data availability

The data that support the findings of this study are available from the corresponding author on reasonable request. Source data are provided with this paper.

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

## Acknowledgements

We are grateful for the financial support provided by Sun Yat-sen University and the China Postdoctoral Science Foundation (2022TQ0380 H.Y.; 2022M723586 H.Y.). We thank Z. Yang for providing instrumental support for fluorescence measurements.

## Author contributions

Z.H. supervised the project. Q.L., H.Y. and Z.H. designed the experiments. Q.L. evaluated the $CO_2$ reduction reactions and performed the photophysical and electrochemical tests. H.Y. investigated the mechanism of $CO_2$ reduction. J.D. carried out the isotopic labeling experiments. M.M. and S.Y. synthesized BIH. Y.C. and J.L. purified **1** and **4**. All authors analyzed the data. Q.L., H.Y. and Z.H. prepared the paper.

## Competing interests

The authors declare no competing interests.
