## [Peer Review File · Nature Communications]

Photocatalytic CO₂ reduction with aminoanthraquinone organic dyesREVIEWER COMMENTS

Reviewer #1 (Remarks to the Author):

Han and co-workers report on the use of aminoanthraquinone organic dyes teamed with an iron complex for the reduction of CO₂ to CO. The work appears to have been carefully performed and sufficient details are included to enable reproduction. The highlight of this work is the high reactivity of the system relative to previously reported systems with organic dyes. Furthermore, interesting conclusions regarding the mechanism are presented that provides new considerations in related mechanistic systems. This work is of broad importance to the community and is acceptable for publication in the current form.

Reviewer #2 (Remarks to the Author):

The results in this paper show an interesting advance and access high activity in the field of photocatalytic CO₂ reduction using anthraquinone coupled to Fe porphyrin as catalyst. Although these studies are not innovated, since there are several studies using Cu and Zn, it is noteworthy that having obtained a high number of TON using a commercial photocatalyst and an Fe porphyrin catalyst. The analysis is clear and detailed on the photophysical and redox potential characteristics as well as the quantification of CO₂. So, it can support the reaction mechanism. Therefore, I consider this article to be recommended for publication.

Dra. Selene Lagunas Rivera

Reviewer #3 (Remarks to the Author):

The manuscript by Han et al. Submitted to Nature Communications describes the use of anthraquinone derivatives as photosensitizers to activate an Iron-based CO₂ catalyst. They report what they believe to be the highest TON to date using organic photosensitizers. I believe that the study is interesting and certainly tries to achieve desirable goals related to CO₂ conversion using non-noble metal systems, but I believe that the present study is not thorough enough and might be too preliminary.

For example, there are too many experimental details missing. The NMR of the different anthraquinone is given, but not the experimental protocols and the corresponding HRMS and IR. If they are all already available from published procedure, then it should be clearly stated and there is no need for the ¹H NMR.

The experimental section should be strengthened. How were the short excited-state lifetime determined? What is the resolution of the instrument and was an IRF used? This is a very important parameter as, in my opinion, most of the results and most of the questions can be assessed via Table 1

and Figure 3. The quenching rate constants are simply too high. The SI only shows a partial part of the emission spectra, which is not very thorough. In addition, given the small excited-state lifetime, the quenching rate constant, if any, will always be very large as the Stern-Volmer relationship requires to divide the slope by the excited-state lifetime to determine the actual quenching rate constant. I mentioned that these quenching rate constants are too large. The quenching rate constant of DMF, solvent used here, should be around $7 \times 10^9 \text{ M}^{-1}\text{s}^{-1}$. Here the authors are almost systematically an order of magnitude higher. This could occur if the species were charged and generated coulombic attraction, or if static quenching was present, but none of these are observed here. Hence this speaks mostly to a larger error in the determination of quenching rate constants. Errors on measurements (k_q , TON and so on) should be added in tables and text.

The author said that a correlation exists between the quenching rate constant and the TON but I would be careful to draw such conclusions. The quenching rate constant represents the first event of electron transfer and does not give any indication about subsequent quenching. If AQH2 is the important product (see below), it is the yield with which this species is produced that is important, and not the first quenching event with BIH.

Thermodynamically, the electron transfer to the iron catalyst is never favorable (as stated by the authors), unless the putative AQH2⁻ species is considered, generated from reduction of AQH2*. How much AQH2 is actually produced during the experiments, and what is the lifetime of that species in the excited state? These parameters should also be considered and discussed.

Did the authors also consider BIH as hydride donor? It is usually a good hydride donor once it has been oxidized by one electron.

Did the authors observe the triplet state of AQ-5? With the Br group, this could lead to increased ISC and generate a triplet state, with a longer excited-state lifetime that would be quenched more efficiently than the other AQ, which could explain the higher yields with that AQ. Indeed, the concentration of 60 mM of BIH, although large, only generated very moderate excited-state quenching, if one looks in the SI. It could be that more quenching of the triplet state is observed.

Regarding the experiments carried out at >500 nm irradiation, did the author calibrate the power and the absorbance? In order to be a relevant comparison, the same amount of excited state must be produced, and hence the absorbance, or the number of absorbed photons, must be controlled.

The author could consider substituting the amine or the phenol with alkyl chains to further substantiate their claim of electron donor/acceptor behavior. It is always very complicated with "free" NH₂ or OH groups to draw straightforward meaningful comparison as they can also be involved in PCET reaction or other equilibria.

More details should be added in the SI and typos should be corrected. For example, it would be interesting to have conditions (solvent etc) in some of the figure captions. The absorbance has not unit

and the "u.a." should hence be deleted for example. This are small changes that would help improve the readability and comprehension of the paper.

Reviewer #1:

“Han and co-workers report on the use of aminoanthraquinone organic dyes teamed with an iron complex for the reduction of CO₂ to CO. The work appears to have been carefully performed and sufficient details are included to enable reproduction. The highlight of this work is the high reactivity of the system relative to previously reported systems with organic dyes. Furthermore, interesting conclusions regarding the mechanism are presented that provides new considerations in related mechanistic systems. This work is of broad importance to the community and is acceptable for publication in the current form.”

We would like to thank Reviewer 1 for his/her helpful comments and critical reading of our manuscript.

Reviewer #2:

“The results in this paper show an interesting advance and access high activity in the field of photocatalytic CO₂ reduction using anthraquinone coupled to Fe porphyrin as catalyst. Although these studies are not innovated, since there are several studies using Cu and Zn, it is noteworthy that having obtained a high number of TON using a commercial photocatalyst and an Fe porphyrin catalyst. The analysis is clear and detailed on the photophysical and redox potential characteristics as well as the quantification of CO₂. So, it can support the reaction mechanism. Therefore, I consider this article to be recommended for publication.”

Dra. Selene Lagunas Rivera

We would like to thank Reviewer 2 for her helpful comments and critical reading of our manuscript.

Reviewer #3:

“The manuscript by Han et al. Submitted to Nature Communications describes the use of anthraquinone derivatives as photosensitizers to activate an Iron-based CO₂ catalyst. They report what they believe to be the highest TON to date using organic photosensitizers. I believe that the study is interesting and certainly tries to achieve desirable goals related to CO₂ conversion using non-noble metal systems, but I believe that the present study is not thorough enough and might be too preliminary.”

We would like to thank Reviewer 3 for his/her insightful reading of our manuscript and for the helpful comments and suggestions. We have taken the referees' advice into account and performed a significant amount of further experiments. We have rewritten the manuscript to have a deeper discussion of the photocatalytic system. Each point is addressed below.

(1) *“For example, there are too many experimental details missing. The NMR of the different anthraquinone is given, but not the experimental protocols and the corresponding HRMS and IR. If they are all already available from published procedure, then it should be clearly stated and there is no need for the ¹H NMR.”*

We thank the referee for the comment. We bought the anthraquinones commercially and purified some of them to ensure no contamination from trace impurities in catalysis. We have added the purification procedure to the revised manuscript and have added the ¹H NMR, ¹³C NMR, IR, and HRMS spectra for all anthraquinones **1-6** to the SI (Supplementary Figures 52-72).

We have also revised the whole experimental sections and all figures/tables (in both MS and SI) to include as much as experimental details.

(2) *“The experimental section should be strengthened. How were the short excited-state lifetime determined? What is the resolution of the instrument and was an IRF used? This is a very important parameter as, in my opinion, most of the results and most of the questions can be assessed via Table 1 and Figure 3. The quenching rate constants are simply too high. The SI only shows a partial part of the emission spectra, which is not very thorough. In addition, given the small excited-state lifetime, the quenching rate constant, if any, will always be very large as the Stern-Volmer relationship requires to divide the slope by the excited-state lifetime to determine the actual quenching rate constant. I mentioned that these quenching rate constants are too large. The quenching rate constant of DMF, solvent used here, should be around $7 \times 10^9 \text{ M}^{-1}\text{s}^{-1}$. Here the authors are almost systematically an order of magnitude higher. This could occur if the species were charged and generated coulombic attraction, or if static quenching was present, but none of these are observed here. Hence this speaks mostly to a larger error in the determination of quenching rate constants. Errors on measurements (k_q , TON and so on) should be added in tables and text.”*

We thank the referee for the comment. We have rewritten the experimental sections to include details for readers. For example, we have added detail procedure for measuring the lifetime of the anthraquinone dye to the manuscript as follows:

“...The excited-state lifetime of the photosensitizer was measured with an FLS 980 fluorescence spectrometer (Edinburgh instruments), in which a picosecond pulsed diode laser ($\lambda = 472 \text{ nm}$ and 406.2 nm) (Edinburgh instruments EPL-470) was used as the excitation source. The λ_{max} emission for each photosensitizer is selected as the emission wavelength. The instrumental response function (IRF) of the instrument was measured using silicon oxide (30 % in H₂O) (Supplementary Fig. 74)...”

Supplementary Figure 74. Emission delay. Emission delay of the picosecond pulsed diode laser ($\lambda = 472$ nm). IRF was measured using silicon oxide (30% in H₂O) in a quartz cuvette (10-mm path length) at 298 K.

The referee brings up a very good point on the quenching experiments. We have investigated potential reactions between BIH and anthraquinones **1-6** by UV-vis and ¹H NMR. We did not see any change of the spectra for each component in the mixtures, which rules out a static quenching mechanism. Please see the following figures. We have also investigated the issue of the unexpected fast fluorescence quenching rate constants. Because the excited-state intramolecular proton transfer of anthraquinone molecule could be affected by the change of proton concentration, addition of BIH (which is slightly basic) will alter the proton concentration and may lead to systematic change of the fluorescence spectra in the excited state. To address the problem, we have re-determined the k_q by measuring the change of the emission lifetime instead of fluorescence intensity. Now the k_q values are all lower than the diffusion-controlled limit. Please see the following data. We have clarified this point in the manuscript as follows:

“Because aminoanthraquinone undergoes excited-state intramolecular proton transfer (ESIPT),⁵⁴ changing proton concentration may affect the fluorescence spectra during quenching experiments. In fact, we observed that addition of BIH (which is slightly basic) to PSs **4** and **5** result in uneven quenching of the fluorescence at different wavelengths and that the fluorescence quenching rate constants (k_q) were calculated to be higher than the diffusion-controlled limit in DMF (Supplementary Figs. 18–19). Since the fluorescence lifetimes of the ESIPT tautomers have been reported to be identical,⁵⁴ we determined the k_q values by measuring the change of fluorescence lifetime in the presence of BIH. The reductive fluorescence quenching of **1-6** were found to be fast near the diffusion-controlled limit ($>10^9$ M⁻¹s⁻¹) (Table 1 and Supplementary Fig. 20).”

Supplementary Figure 23. ¹H NMR spectra. ¹H NMR (400 MHz, 298 K) spectra of BIH, PS 1, and a mixture of BIH and PS 1 in *d*₆-DMSO in air.

Supplementary Figure 24. ¹H NMR spectra. ¹H NMR (400 MHz, 298 K) spectra of BIH, PS 2, and a mixture of BIH and PS 2 in *d*₆-DMSO in air.

Supplementary Figure 25. ¹H NMR spectra. ¹H NMR (400 MHz, 298 K) spectra of BIH, PS 3, and a mixture of BIH and PS 3 in *d*₆-DMSO in air.

Supplementary Figure 26. ¹H NMR spectra. ¹H NMR (400 MHz, 298 K) spectra of BIH, PS 4, and a mixture of BIH and PS 4 in *d*₆-DMSO in air.

Supplementary Figure 27. ^1H NMR spectra. ^1H NMR (400 MHz, 298 K) spectra of BIH, PS 5, and a mixture of BIH and PS 5 in d_6 -DMSO under CO_2 .

Supplementary Figure 28. ^1H NMR spectra. ^1H NMR (400 MHz, 298 K) spectra of BIH, PS 6, and a mixture of BIH and PS 6 in d_6 -DMSO in air.

Supplementary Figure 18. Fluorescence emission quenching. Stern-Volmer plots at 620 nm and 647 nm (right) from fluorescence quenching (left) ($\lambda_{\text{exc}} = 530$ nm) of **4** (50 μM) by BIH in DMF in a quartz cuvette (10-mm path length) at 298 K under N_2 . Source data are provided as a Source Data file.

Supplementary Figure 19. Fluorescence emission quenching. Stern-Volmer plot at 635 nm and 648 nm (right) from fluorescence quenching (left) ($\lambda_{\text{exc}} = 530$ nm) of **5** (50 μM) by BIH in DMF in a quartz cuvette (10-mm path length) at 298 K under CO_2 . Source data are provided as a Source Data file.

Supplementary Figure 20. Fluorescence lifetime quenching. Stern-Volmer plots of fluorescence lifetime quenching of 50 μM (a) **1** ($\lambda_{\text{em}} = 600 \text{ nm}$); (b) **2** ($\lambda_{\text{em}} = 650 \text{ nm}$); (c) **3** ($\lambda_{\text{em}} = 662 \text{ nm}$); (d) **4** ($\lambda_{\text{em}} = 620 \text{ nm}$); (e) **5** ($\lambda_{\text{em}} = 635 \text{ nm}$); and (f) **6** ($\lambda_{\text{em}} = 607 \text{ nm}$) in DMF in a quartz cuvette (10-mm path length) under N_2 (for **1–4** and **6**) or under CO_2 (for **5**) at 298 K. The error bars denote standard deviations, based on 3 separated runs. The excitation wavelength is 472 nm. Source data are provided as a Source Data file.

We have also determined the errors for TON and TOF based on at least three separated experiments. The k_q values were determined from fitting the series of lifetimes (each point obtained from averaging 3 separated runs) (please see the last figure). We have added these results to Table 1 as follows:

Table 1. Photophysical, electrochemical, and photocatalytic CO₂ reduction data of PSs 1–6.

PS	$\lambda_{\text{max}}^{\text{abs}}/\text{nm}$ ($\epsilon \text{ M}^{-1} \text{ cm}^{-1}$)	$\lambda_{\text{max}}^{\text{em}}/\text{nm}$	E_{red} (V vs. SCE)	TON _{Fe^a}	TOF _{Fe^a}	Sel _{CO^a} (%)	CO ^b (μmol)	TON _{PS^b}	Φ^c (%)	Φ_{FL}^d (%)	τ_0^d (ns)	k_q ($\text{M}^{-1}\text{s}^{-1}$)
1	478 (6790)	600	-0.96, -1.59	2395 ± 228	1510 ± 104	99.6 ± 0.1	50 ± 6	2011 ± 257	8.9 ± 0.8	4.7	0.88	3.9 × 10 ⁹
2	528 (8940)	650	-1.10, -1.70	2738 ± 190	69 ± 8	99.5 ± 0.2	12 ± 2	482 ± 76	0.3 ± 0.04	2.3	0.72	2.1 × 10 ⁹
3	592 (15810)	662	-1.15, -1.64	3551 ± 501	593 ± 24	99.3 ± 0.2	38 ± 3	1523 ± 126	3.0 ± 0.1	5.3	0.82	2.7 × 10 ⁹
4	532 (12040)	620	-0.84, -1.44	8360 ± 449	1614 ± 112	99.6 ± 0.1	71 ± 4	2849 ± 161	8.1 ± 0.3	7.1	0.94	5.2 × 10 ⁹
5	534 (9170)	635	-0.68, -1.19	21616 ± 2351	4028 ± 669	> 99.9	153 ± 10	6012 ± 606	11.1 ± 0.9	7.0	1.02 ^e	7.5 × 10 ⁹
6	490 (7460)	607	-0.86, -1.30	907 ± 154	93 ± 17	98.5 ± 0.6	30 ± 2	1183 ± 78	2.0 ± 0.3	2.6	0.66	1.6 × 10 ⁹

^a 60 mM BIH, 0.6 μM FeTDHPP, and 20 μM PS, $\lambda > 400 \text{ nm}$. TON and Sel_{CO} calculated in 48 h, TOF calculated in 0.5 h for PS 1, 2 h for PS 2, and 1 h for PSs 3–6. ^b 60 mM BIH, 20 μM FeTDHPP and 5 μM PS, $\lambda > 400 \text{ nm}$, amount of CO and TON_{PS} calculated in 72 h. ^c 60 mM BIH, 20 μM FeTDHPP and 20 μM PS, $\lambda = 450 \text{ nm}$, Φ calculated in 1 h. ^d 50 μM PS, a picosecond pulsed diode laser ($\lambda = 472 \text{ nm}$) was used as the excitation source. ^e Under CO₂. The $\lambda_{\text{max}}^{\text{em}}$ of each photosensitizer is selected as the emission wavelength. Error bars denote standard deviations, based on at least three separated runs. Source data are provided as a Source Data file.

(3) “The author said that a correlation exists between the quenching rate constant and the TON but I would be careful to draw such conclusions. The quenching rate constant represents the first event of electron transfer and does not give any indication about subsequent quenching. If AQH₂ is the important product (see below), it is the yield with which this species is produced that is important, and not the first quenching event with BIH.”

We thank the referee for the comment. We agree with the referee that the first quenching step may not reflect the subsequent reduction of anthraquinone. However, with the more accurate k_q values, the correlation between the first quenching rate constant and the TOF becomes more pronounced (please see the following figure). We think this observation is interesting for providing a hint in understanding the mechanism for future studies. We have modified the text to make this point clearer for readers as follows:

“The observation of the PSH₂⁻ species by UV–vis during CO production suggests that formation of PSH₂⁻ may be rate-limiting in catalysis. Interestingly, we found that the TOF of CO production and the first reductive quenching k_q of PSs 1–6 follows a generally linear trend, in which a faster quenching rate is observed with a higher TOF (Fig. 6). However, further evidence is necessary to identify the substituent effect on the photo-conversion of PSH₂ to PSH₂⁻ in CO₂ reduction.”

Fig. 6 Relationship of TOF and k_q . Plot of of the initial rates of CO generation with respect to the reductive quenching rate constants of PSs 1–6. The dotted line is shown for viewing convenience. Error bars denote standard deviations, based on at least three separated runs. Source data are provided as a Source Data file.

(4) “Thermodynamically, the electron transfer to the iron catalyst is never favorable (as stated by the authors), unless the putative AQH_2^- species is considered, generated from reduction of AQH_2^* . How much AQH_2 is actually produced during the experiments, and what is the lifetime of that species in the excited state? These parameters should also be considered and discussed.”

We thank the referee for the question. We have performed further experiments to investigate this. We regret that our previous assignment of the ~ 400 nm species as an AQH_2 is incorrect. Based on our new results, this species observed in UV-vis spectra is likely the AQH_2^- species.

Because of potential inversions of the semiquinone and hydroquinone species (*J. Am. Chem. Soc.* **2017**, *139*, 5225-5232), their reductions are at the same potential, which is thermodynamically infeasible to generate the $Fe(0)$ species in catalysis. Thus, addition of the FeTDHPP catalyst to the AQH_2 should not generate any CO. However, we observed the generation of CO and Fe^I by adding FeTDHPP to a solution previously subjected to light irradiation until the generation of the 400 nm species (please see the following figure). Based on the proposed mechanism, the maximum yield of CO should be 0.33 equivalence of the photosensitizer. In the experiments, we observed an average of 0.34 equivalence of CO based on two separated runs. Thus, the conversion of this species at ~ 400 nm to generate Fe^0 and then CO is almost in a quantitative yield. In a control experiment before generating the 400 nm species, we observed no CO. Instead, the Fe^{II} and Fe^I species were observed and the anthraquinone **5** was recovered. Thus,

we conclude that this species at ~ 400 nm is an AQH_2^- . We have revised the manuscript to include the results as follows:

“The intermediate generated from AQ at ~ 400 nm starts to appear at 15 min and continue to increase in 2 h during photolysis (Fig. 4b), which is commonly observed with PSs 1–6 during CO_2 reduction (Supplementary Figs. 31–36). Although this intermediate exhibits similar absorption feature as the AQH_2 , much slower generation of the species than that for the $\text{AQH}\cdot$ suggests it is not a AQH_2 . Furthermore, we observed fast generation of CO and an Fe(I) species by adding 0.25 equiv of FeTDHPP (with respect to PS) to the 406 nm species generated from light irradiation of a mixture of **5** and BIH (Supplementary Fig. 42). In the experiments, an average of 0.34 equiv of CO (vs PS) was obtained, which is close to the theoretical maximum yield (0.33 equiv) based on the proposed mechanism in Scheme 1. In a control experiment before generating the 406 nm species, no CO was detected (Supplementary Fig. 43). Based on these results, this photoproduct at ~ 400 nm is tentatively assigned to a PSH_2^- .”

Supplementary Figure S42. UV-vis absorption spectra. UV-vis absorption spectra of systems containing PS **5** (40 μM) and BIH (20 mM) in DMF (2 mL) in a quartz cuvette (10-mm path length) under CO_2 upon irradiation with white LED light ($\lambda > 400$ nm, 100 mW/cm^2) for 20 min (black), and then added with FeTDHPP (0.25 equiv vs **5**) for 1 min (red). The amount of CO quantitated by GC was $0.027 \pm 0.001 \mu\text{mol}$ after addition of FeTDHPP.

Supplementary Figure S43. UV-vis absorption spectra. UV-vis absorption spectra of systems containing PS **5** (40 μM) and BIH (20 mM) in DMF (2 mL) in a quartz cuvette (10-mm path length) under CO_2 before (black) and after irradiation with white LED light ($\lambda > 400 \text{ nm}$, 100 mW/cm^2) for 5 minutes (red), and then added with FeTDHPP (0.25 equiv vs **5**) for 1 min (blue). No CO was detected after adding FeTDHPP by GC analysis.

We have also measured the lifetime of the AQH_2^* . After reducing the AQ in NaBH_4 and then adding acetic acid, we are able to obtain the fluorescing AQH_2 species, which gives a UV-vis spectrum in agreement with a previous report (*Chem. Commun.* **2016**, 52, 1210-1213). The lifetime of the species was determined to be 17.3 ns in DMF and 18.1 ns in DMSO. Thus, the 17 μs reported in the Chem. Comm. paper is probably a typo. The quenching rate constant of the AQH_2 by BIH is measured to be $6.07 \times 10^9 \text{ M}^{-1}\text{s}^{-1}$. We have tried to perform the same reaction with aminoanthraquinones. However, isolation of spectrally pure PSH_2 species is not successful, which is probably due to that NaBH_4 is not an appropriate reductant in the reaction. We have revised the manuscript by including the figures to the SI and adding the following sentences to the manuscript:

“Indeed, the AQH_2 species generated from subsequent reductions and protonations gives a relatively long-lived (17.3 ns) fluorescence (Supplementary Figs. 39–40), which can be quickly quenched by BIH (Supplementary Fig. 41). Moreover, the AQH_2 moiety in a pentad system has been reported to be highly fluorescent with a long lifetime of 4.7 μs .⁵⁹ Thus, a plausible photochemical pathway involves reduction of the Fe catalyst by a PSH_2^- photoproduct, which can be generated from excitation of the PSH_2 followed by reductive quenching (Scheme 1).”

Supplementary Figure S39. Spectra of AQH₂. UV-vis spectrum (a), excitation spectrum (b) and emission spectrum (c) of AQH₂ in DMSO-d₆ in a quartz cuvette (10–mm path length). Similar results have been previously reported.¹ The AQH₂ species was generated from a procedure as follows: A DMSO-d₆ solution (1.0 mL) containing 0.1 mmol AQ was added to NaBH₄ (0.15 mmol) under N₂. The reaction mixture was allowed to stir for 6 h at room temperature under N₂. Then 1000 eq. CH₃COOH was added to generate the AQH₂. The spectra were recorded in a diluted solution (dilution factor of 2000) under N₂.

Supplementary Figure S40. Emission decay of AQH₂. Emission decay of the *in situ* generated AQH₂ in DMF (a) or DMSO-d₆ (b) in a quartz cuvette (10–mm path length) under N₂ at 298 K. The lines were fitted with a single exponential. The AQH₂ species

was generated from a procedure as follows: A DMSO-d₆ solution (1.0 mL) containing 0.1 mmol AQ was added to NaBH₄ (0.15 mmol) under N₂. The reaction mixture was allowed to stir for 6 h at room temperature under N₂. Then 1000 eq. CH₃COOH was added to generate the AQH₂. The spectra were recorded in a diluted solution (dilution factor of 2000) under N₂.

Supplementary Figure S41. Luminescence lifetime quenching of AQH₂. Emission decay of the *in situ* generated AQH₂ with addition of different concentrations of BIH in DMF in a quartz cuvette (10-mm path length) under N₂ at 298 K, with lines fitted with a single exponential (a). A Stern-Volmer plot showing a quenching rate constant of $6.07 \times 10^9 \text{ M}^{-1} \cdot \text{s}^{-1}$ (b). The excitation wavelength was 406.5 nm. The AQH₂ species was generated from a procedure as follows: A DMSO-d₆ solution (1.0 mL) containing 0.1 mmol AQ was added to NaBH₄ (0.15 mmol) under N₂. The reaction mixture was allowed to stir for 6 h at room temperature under N₂. Then 1000 eq. CH₃COOH was added to generate the AQH₂. The spectra were recorded in a diluted solution (dilution factor of 2000) under N₂.

(5) “Did the authors also consider BIH as hydride donor? It is usually a good hydride donor once it has been oxidized by one electron.”

We thank the referee for the question. BIH has been studied as a strong organic hydride donor and also an one-electron reducing agent (*J. Am. Chem. Soc.* **2008**, *130*, 2501-2516). However, to the best of our knowledge, no direct reduction of CO₂ by this type of organic hydride donor has been reported so far (*Organometallics* **2018**, *37*, 3972–3982). In our studies, we have not observed any formate from CO₂ reduction when using BIH as the donor, which suggests that BIH is probably not acting as a hydride donor in CO₂ reductions. We think it is more likely that BIH works as a two-electron and one proton donor in CO₂ reduction as described by Ishitani and co-workers (*ACS Catal.* **2017**, *7*, 3394–3409) in the following figure. However, we do observe the generation of AQH• from the UV-vis spectra (Figure 4). Thus, BIH may actually act as a hydride donor during anthraquinone reduction.

(6) “Did the authors observe the triplet state of AQ-5? With the Br group, this could lead to increased ISC and generate a triplet state, with a longer excited-state lifetime that would be quenched more efficiently than the other AQ, which could explain the higher yields with that AQ. Indeed, the concentration of 60 mM of BIH, although large, only generated very moderate excited-state quenching, if one looks in the SI. It could be that more quenching of the triplet state is observed.”

We thank the referee for the question. We have measured the fluorescence quantum yields (QYs) for anthraquinones **1-6** (please see Table 1). The fluorescence QYs are in the same level as the QYs in CO production for most of the anthraquinones, although we should be really looking at the fluorescence of the hydroquinone species. Based on the study of a simple anthraquinone (please see our reply to question 4), the hydroquinone species was also found to be highly fluorescent. This results along with the correlation between the TOF and the quenching rate constant (please see our reply to question 3) suggest that it is reasonable to consider that electron transfer from BIH to anthraquinone occur through fluorescence quenching. In addition, the triplet QYs of **1** and **3** have been reported to be 3% and <0.1% in MeOH (*J. Photochem. Photobiol., A* **1988**, *41*, 227-244). Because compound **5** could have a higher triple QY because of its heavy Br atom, we think electron transfer through the triple state of anthraquinone is another plausible photochemical pathway, especially for **1** and **5**. We have revised the manuscript to describe possible influence of the triple state on electron transfer as follows:

“However, since the triplet quantum yields of PSs **1** and **3** have been reported to be 3% and < 0.1% in methanol respectively,⁵⁵ it should be noted that reductive quenching occurring through ³PS* is another plausible photochemical pathway, especially for the heavy-atom containing compound **5**.”

(7) “Regarding the experiments carried out at > 500 nm irradiation, did the author calibrate the power and the absorbance? In order to be a relevant comparison, the same amount of excited state must be produced, and hence the absorbance, or the number of absorbed photons, must be controlled.

We thank the referee for the comment. We have performed UV-vis experiments using blue LED light ($\lambda = 450$ nm) and green LED light ($\lambda = 525$ nm) and adjusted their light powers to get a similar number of photons. The results are consistent with our previous

observation that the Fe^{I} was only observed under the 450 nm light irradiation. Please see the following Figures. We have included these Figures to the SI and added the following sentences to the MS to describe the results:

“Similar results are observed in experiments performed under 450 nm and 525 nm LED light with similar photon numbers (Supplementary Figs. 49–50). Meanwhile, the production of CO under 525 nm LED light (TON = 1.4 in 8 h) is also lower than that under the 450 nm LED light (TON = 344 in 8 h).”

Supplementary Figure 49. UV-vis absorption spectra. Systems containing 30 mM BIH, 20 μM FeTDHPP, and 20 μM **5** in CO_2 -saturated DMF in a quartz cuvette (10-mm path length) at 298 K under irradiation with a blue LED ($\lambda = 450$ nm, $\Delta P \cdot \lambda = 12300$ $\text{mW} \cdot \text{nm}/\text{cm}^2$). Irradiation time ranging from 0 to 30 min (a), and from 30 min to 8 h (b).

Supplementary Figure 50. UV-vis absorption spectra. Systems containing 30 mM BIH, 20 μM FeTDHPP, and 20 μM **5** in CO_2 -saturated DMF in a quartz cuvette (10-mm path length) at 298 K under irradiation with a green LED ($\lambda = 525$ nm, $\Delta P \cdot \lambda = 13203$ $\text{mW} \cdot \text{nm}/\text{cm}^2$). Irradiation time ranging from 0 to 30 min (a), and from 30 min to 8 h (b).

(8) “The author could consider substituting the amine or the phenol with alkyl chains to further substantiate their claim of electron donor/acceptor behavior. It is always very complicated with “free” NH_2 or OH groups to draw straightforward meaningful comparison as they can also be involved in PCET reaction or other equilibria.”

We thank the referee for the suggestion. We have carefully compared the reduction peaks of the anthraquinones, including the one with a methyl substituent at the 2-position. The SWV data clearly show that the reduction potentials shift to different directions with the $-\text{OH}$ and $-\text{NH}_2$ substituents. Please see the following figure, we have added the figure to the SI and have revised the statement in the manuscript as follows:

“Based on the electrochemical data of **1–6** and 1-amino-2-methylantraquinone (Table 1 and Supplementary Figs. 9–10), the $-\text{NH}_2$ group acts an electron-donating group while the $-\text{OH}$ group is electron-withdrawing on AQ.”

Supplementary Figure 10. Electrochemical study. Square wave voltammetry of 1.0 mM (a) **5**; (b) **6**; (c) **4**; (d) **1**; (e) 1-amino-2-methylantraquinone; (f) **3**; and (g) **2** in 5 mL DMF containing 0.1 M TBAPF_6 under N_2 . Experiments were conducted using a glassy carbon working electrode (3.0 mm in diameter), a Pt wire counter electrode, and a KCl-saturated calomel electrode at a scan rate of $100 \text{ mV} \cdot \text{s}^{-1}$. Source data are provided as a Source Data file.

(9) “More details should be added in the SI and typos should be corrected. For example, it would be interesting to have conditions (solvent etc) in some of the figure captions. The absorbance has not unit and the "u.a." should hence be deleted for example. This are small changes that would help improve the readability and comprehension of the paper.”

We thank the referee for the comments. As mentioned in our reply to your questions 1 and 2, we have completely revised the experimental details in both the MS and the SI.

REVIEWERS' COMMENTS

Reviewer #3 (Remarks to the Author):

It is the second time that I've had the opportunity to reverse this manuscript. The other two reviewers were very positive about the work and recommended publication "as is" whereas I suggested several experiments, controls and verification. The authors have answered all my queries, have performed numerous judicious experiments and have greatly improve the manuscript. They provided response and argumentation to all my comments and have justifiably argued the changes that they performed. I would therefore recommend acceptance of this manuscript.

As a side note, the authors have very transparently provided a source data file. I looked at the data for Figure 18, where the spectral shape shifts with excited-state quenching. If the authors were to integrate the total intensity, rather than the intensity at two wavelength, they would get a much better Stern-Volmer analysis, with $R^2=0.99$ and a corresponding quenching rate constant of $9.1 \times 10^9 \text{ M}^{-1}\text{s}^{-1}$. In value, this is different from the $5.2 \times 10^9 \text{ M}^{-1}\text{s}^{-1}$ obtained by excited-state lifetime measurements but in practice I believe that these two values are the same but just speak towards the large errors (not reported on k_q in the revised manuscript). associated with performing measurements with compounds with such short excited-state lifetimes.

Reviewer #3:

“It is the second time that I've had the opportunity to reverse this manuscript. The other two reviewers were very positive about the work and recommended publication "as is" whereas I suggested several experiments, controls and verification. The authors have answered all my queries, have performed numerous judicious experiments and have greatly improve the manuscript. They provided response and argumentation to all my comments and have justifiably argued the changes that they performed. I would therefore recommend acceptance of this manuscript.”

We would like to thank Reviewer 3 for his/her helpful comments and critical reading of our manuscript, which enabled us to improve the clarity and quality of our work.

“As a side note, the authors have very transparently provided a source data file. I looked at the data for Figure 18, where the spectral shape shifts with excited-state quenching. If the authors were to integrate the total intensity, rather than the intensity at two wavelength, they would get a much better Stern-Volmer analysis, with $R^2=0.99$ and a corresponding quenching rate constant of $9.1 \times 10^9 \text{ M}^{-1}\text{s}^{-1}$. In value, this is different from the $5.2 \times 10^9 \text{ M}^{-1}\text{s}^{-1}$ obtained by excited-state lifetime measurements but in practice I believe that these two values are the same but just speak towards the large errors (not reported on k_q in the revised manuscript). associated with performing measurements with compounds with such short excited-state lifetimes.”

We thank the referee for the comment. We have recalculated the k_q based on the peak areas of the fluorescence spectra. However, some of the values are still much higher than the diffusion-controlled limit. Please see the following Figure for example. As we have mentioned in our last response, because of the excited-state intramolecular proton transfer of aminoanthraquinone, the change of proton concentration may significantly affect the fluorescence spectra. Thus, a reliable quenching rate constant cannot be determined from fluorescence quenching experiments. In fact, we have encountered the same problem from our other aminoanthraquinone systems. However, we are able to get reasonable k_q values from measuring the change of the fluorescence lifetime (please see one of our recent reports: *J. Am. Chem. Soc.* **2022**, *144*, 19680).

Figure R1. Fluorescence emission quenching. The k_q was calculated from the Stern-Volmer plot based on the ratio of the peak area (550 nm to 800 nm) from fluorescence quenching (left) ($\lambda_{exc} = 530 \text{ nm}$) of **5** ($50 \mu\text{M}$) by BIH in DMF in a quartz cuvette (10–mm path length) at 298 K under CO_2 . S_0 and S represent the peak areas of fluorescence intensities of the photosensitizer in the absence and presence of a quencher, respectively.